# Robotic Rectal Cancer Surgery: Perioperative and Long-Term Oncological Outcomes of a Single-Center Analysis Compared with Laparoscopic and Open Approach

**DOI:** 10.3390/cancers17050859

**Published:** 2025-03-02

**Authors:** Shachar Laks, Michael Goldenshluger, Alexander Lebedeyev, Yasmin Anderson, Ofir Gruper, Lior Segev

**Affiliations:** 1Faculty of medicine, Tel-Aviv University, Tel-Aviv 6997801, Israel; shacharl@wmc.gov.il (S.L.); goldenm9@ccf.org (M.G.); anderson@mail.tau.ac.il (Y.A.); ofirgruper@mail.tau.ac.il (O.G.); 2Department of Surgery, Wolfson Medical Center, Holon 5822012, Israel; 3Division of Surgery, The Chaim Sheba Medical Center, Tel-Hashomer 5266202, Israel; alex.lebdev@sheba.health.gov.il

**Keywords:** rectal cancer, robotic surgery, laparoscopy, survival, recurrence

## Abstract

Robotic-assisted surgery is a promising option with distinct advantages in rectal cancer surgery, which were shown mainly regarding the immediate postoperative outcomes such as decreasing the length of hospital stay, lowering wound infection rates, readmission rates, conversions rates, and even improving the wellbeing and work ergonomics of the operating surgeon. However, the evidence concerning the efficacy of the robotic approach in rectal cancer clearance and long-term oncological outcomes is still conflicting, and therefore, the optimal surgical approach for rectal cancer is still questionable. Hence, this study aimed to compare the short- and long-term outcomes of robotic-assisted rectal cancer surgery with the conventional laparoscopic-assisted approach and the open approach.

## 1. Introduction

Despite evolving oncological neoadjuvant and adjuvant treatments, surgery still remains the mainstay curative option for patients with rectal cancer, and total mesorectal excision (TME) is the gold standard technique in rectal cancer surgery, as it provides improved control of local recurrence and overall survival [1]. Although it has been traditionally approached through open laparotomy, minimally invasive approaches are increasingly utilized in patients with rectal cancer, and randomized clinical trials including the COLOR II and the COREAN trials have demonstrated laparoscopic rectal cancer surgery to have similar or better short-term surgical outcomes and similar long-term oncological outcomes compared with open rectal cancer surgery [2,3,4]. However, laparoscopic rectal cancer surgery might be very challenging due to the confines of the narrow bony pelvis and the restricted flexibility of the rigid laparoscopic arms, hence raising concerns regarding tumor clearance and controlling distal and circumferential resection margins. The recent ACOSOG Z6051 and ALaCaRT trials failed to demonstrate the non-inferiority of laparoscopy compared with open surgery concerning pathological outcomes, questioning the true oncological safety of laparoscopic surgery for rectal cancer [5,6]. Those limitations of the laparoscopic platform could be potentially overcome by the robotic approach with its articulating instruments, three-dimensional depth of field view, stable camera platform, and possibly more precise tissue dissection [7]. Robotic-assisted TME was first described in 2006 [8], and since then, different retrospective studies compared between the robotic and laparoscopic surgical approach, and the majority of them showed no significant differences in postoperative morbidity and oncological outcomes [9,10,11]. The ROLARR randomized clinical trial comparing robotic-assisted surgery with laparoscopic surgery for rectal cancer failed to show clear superiority of the robotic approach in terms of conversion rates [12], and the COLARAR randomized trial concluded that robotic-assisted surgery did not significantly improve the TME quality compared with conventional laparoscopic surgery [13]. Despite the growing utilization of robotic surgery in rectal cancer treatment, the question of whether it offers substantial clinical benefits over the more traditional laparoscopic or open approaches remains unanswered. A recent meta-analysis comparing the robotic and laparoscopic approach for rectal cancers found no significant difference in overall survival and postoperative complications between the groups [14]. Most of the previous literature comparing rectal cancer surgery approach outcomes was limited to pairwise comparison only, and just a few studies have compared the open, laparoscopic, and robotic approaches together. Therefore, the aim of this study was to present a contemporary large cohort long-term comprehensive analysis of robotic rectal cancer surgery compared with laparoscopic and open approach outcomes.

## 2. Materials and Methods

**Study design/population:** This was a single-center retrospective study. After the approval of the institutional review board, we searched our electronic medical records to extract all patients who underwent radical rectal resection between the years 2010 and 2020 using all appropriate surgery codes including anterior resection of rectum, proctectomy with coloanal anastomosis, and abdomino-perineal resection (APR). From this database, we chose only patients with a diagnosis of primary non-metastatic rectal adenocarcinoma, designated clinically as stage I-III according to the American Joint Committee on Cancer Guidelines 7th edition [15]. We excluded non-elective, emergency cases, and palliative procedures. The final cohort was divided into three groups according to the primary surgical approach: robotic (Da Vinci Si or Xi Surgical System Intuitive Surgical, Sunnyvale, CA, USA), laparoscopic, or open approach. The decision regarding the surgical approach was taken according to the surgeon’s discretion. Intraoperative conversion to a different surgical approach was analyzed as the original intention to treat. **Study variables and outcome measures:** Patients’ characteristics and demographics included age, gender, body mass index (BMI), smoking status, personal surgical history, comorbidities, and American Society of Anesthesiologists score (ASA). Preoperative disease characteristics included presenting symptoms, disease work-up and imaging, tumor location in the rectum (lower rectum ≤ 5 cm, mid rectum 5–10 cm, upper rectum > 10 cm), pretreatment clinical staging, neoadjuvant oncological therapy, and preoperative laboratory values. Operative characteristics included the extent of resection performed, such as abdomino-perineal resection (APR), low anterior resection of rectum (with TME), and anterior resection of rectum (with tumor-specific TME), type of stoma constructed (which was performed at the discretion of the surgeon), additional procedure undertaken during surgery, type of anastomosis, skin incision location, conversion to open procedure, and intraoperative complications. Postoperative surgical outcomes included length of hospital stay (LOS), 30-day readmission rates and in hospital or 30-day postoperative complications, and their severity grading according to the Clavien–Dindo classification system [16]. A Clavien–Dindo grade > 2 was defined as a major complication. Histopathological surgical results included tumor size, differentiation, pathological stage, number of harvested lymph nodes, and distance of tumor from distal margins. Long-term oncological outcomes included details about adjuvant radiation therapy or adjuvant chemotherapy, the presence of local or distant disease recurrence, and mortality data. Recurrence-free survival (RFS) was defined as the period from the date of surgery to the date of the first recurrence. If tumor recurrence was not recorded, RFS was defined as the time between the date of surgery and the date of the last follow-up. Overall survival (OS) was calculated from the date of surgery to either the date of death or the date of the last follow-up visit. **Statistical analysis:** For numerical variables, we reported the median and range for each group and tested their difference using the Wilcoxon rank test. For categorical variables, we used the chi-squared test of independence using bootstrap to obtain a more accurate *p*-value when there were categories with low frequencies. *p* value < 0.05 was considered statistically significant. Univariate and multivariate analyses for risk factors for major complications were performed using the logistic regression model. The multivariate analysis was adjusted for the covariables which had a *p* < 0.3 in the univariate analysis (age, gender, tumor distance from anal verge, surgical approach, neoadjuvant radiation, and stoma creation). We used the Kaplan–Meier procedure for the estimated survival curves and their 95% confidence intervals. Univariate and multivariate analyses for disease recurrence were based on the Cox proportional hazard regression. The multivariate analysis for disease recurrence was adjusted for the covariables which had a *p* < 0.3 in the univariate analysis (age, surgical approach, tumor distance from anal verge, and positive margins).

## 3. Results

### 3.1. Patients Characteristics and Demographics

The cohort included 526 patients with a median age of 64 years (range 31–89), of whom 103 patients were in the robotic group, 144 in the open group, and 279 patients in the laparoscopic group. All three groups were similar in relation to their demographics and baseline characteristics including comorbidities, ASA score, and prior abdominal operations (Table 1).

### 3.2. Disease Presentation and Preoperative Work-Up

The robotic group had significantly more patients with lower rectum tumors compared with the other groups (24.3% versus 12.7% in the open group and 6% in the laparoscopic group, *p* < 0.001) and significantly fewer patients with upper rectal tumors (38.8% versus 49.3% and 66.5%, respectively, *p* < 0.001). Consequently, the median tumor distance from the anal verge was the lowest among the robotic group (8 cm versus 9 cm and 12 cm, respectively, *p* < 0.001). The robotic group had significantly more patients with locally advanced tumors as could be expressed by their higher rates of clinical stage 3 tumors (65.6% versus 51.2% and 50.2%, respectively, *p* = 0.004) and their lower rates of clinical stage 1 tumors (11.5% versus 22.5% and 30.5%, respectively, *p* = 0.004). Accordingly, the rates of neoadjuvant radiotherapy (long or short course) were significantly higher among the robotic group (70.9% versus 54.2% and 39.5%, respectively, *p* < 0.001) (Table 2).

### 3.3. Surgical Procedure

APR was significantly more prevalent among the robotic group (17.6% versus 2.1% and 4.3%, respectively, *p* < 0.001). Consequently, the robotic group had higher rates of a permanent end colostomy compared with the open and laparoscopic group (18.4% versus 7.7% and 7.8%, respectively, *p* < 0.001). An additional surgical intervention during surgery was much more prevalent among the open group, and salpingo-ophorectomy was the most common of these (19.7% in the open group compared with 1.9% in the robotic group and 3.6% in the laparoscopic group, *p* < 0.001). With regard to the anastomosis, a hand-sewn coloanal anastomosis was significantly more prevalent in the open group (12.9% compared with 3.6% in the robotic group and 4.2% in the laparoscopic group, *p* < 0.001). Conversion rates to an open laparotomy were significantly higher among the laparoscopic group compared to the robotic group (23.1% versus 6.8%, respectively, *p* = 0.001). Intraoperative complications were significantly more prevalent within the open group (23.2% compared with 10.7% among the robotic group and 13.5% among the laparoscopic group, *p* = 0.011) (Table 3).

### 3.4. Postoperative Surgical Outcomes

The open group patients had significantly longer hospital stay (LOS) compared with the other groups (10 days compared with 7 and 8 days among the robotic and laparoscopic groups, respectively, *p* < 0.001). The postoperative overall complication rate was significantly higher among the open group (76% compared with 68.9% among the robotic group and 59.1% among the laparoscopic group, *p* = 0.002), and surgical site infection was the main morbidity differentiating the open groups from the other groups (28.9% compared with 7.8% and 10.5%, respectively, *p* < 0.001). Additionally, major complications were also more prevalent among the open group (23.9% versus 13.6% in the robotic group and 12.8% in the laparoscopic group, *p* = 0.01) (Table 4).

A logistic regression model to test for risk factors for major complications found male gender (OR, 1.676; 95% CI, 1.004–2.798; *p* = 0.048), open approach (OR, 1.959; 95% CI, 1.132–3.390; *p* = 0.026), and loop ileostomy construction (OR, 3.416; 95% CI, 1.570–7.431; *p* = 0.007) to be significantly associated with postoperative major complications both in univariate and multivariate analyses. Shorter tumor distance from the anal verge was associated with major complications in univariate analysis but not after multivariate analysis. The robotic and laparoscopic approach, age, BMI, smoking, and preoperative radiation were not found to be associated with major complications (Table 5).

### 3.5. Histopathological Results

The three groups were similar in the rates of complete pathological response, in the pathological stage distribution, number of harvested lymph nodes, lympho-vascular invasion rate, perineural invasion, tumor differentiation, mucinous tumors, and signet-ring cell features.

The distal margin was significantly larger among the robotic group (3.5 cm compared with 1.9 cm in the open group and 2.5 cm in the laparoscopic group, *p* = 0.006). The rate of involved margins (distal or radial margins) was 2.1% (11 patients) of the entire cohort, with no significant differences between the groups (Table 6).

### 3.6. Long-Term Oncological Outcomes

After a median follow-up time of 59 months (range of 1–171 months), no significant differences were noted between the groups in overall survival (OS) and recurrence-free survival (RFS) (Table 7, Figure 1).

The 5-year overall survival in the robotic group was 92.3% compared with 90.5% and 88.3% in the laparoscopic and open groups, respectively (*p* = 0.12). The 5-year disease-free survival in the robotic group was 68% compared with 71% and 63% in the laparoscopic and open groups, respectively (*p* = 0.2). In addition, there were no differences between the groups in OS and RFS after stratifying the cohort by clinical stage (Figure 2).

The Cox regression model to asses for risk factors for disease recurrence has found shorter tumor distance from anal verge (OR, 0.954; 95% CI, 0.914–0.996; *p* = 0.034) and involved distal/radial surgical margins (OR, 3.599; 95% CI, 1.320–9.812; *p* = 0.037) to be significantly associated with disease recurrence both in univariate and multivariate analyses. The surgical approach was not found to be associated with disease recurrence (Table 8).

### 3.7. Subgroup Analysis

In order to minimize the bias related to heterogeneous patients included in our cohort, we have conducted a subgroup analysis including only patients with mid and low rectal cancer who have received neoadjuvant radiation therapy. This analysis included 197 patients (82 lap, 57 open, and 58 robot), and the three groups were similar in terms of demographics, disease presentation, and preoperative work-up (including tumor location and clinical stage distribution) (Appendix A). There were still no significant differences between the groups in OS and DFS. The 5-year DFS in the robotic group was 66.9% compared with 69% and 65% in the laparoscopic and open groups, *p* = 0.92 (Appendix A). An additional subgroup analysis including only patients with upper rectal cancer was performed consisting of 297 patients (186 lap, 71 open, and 40 robot) (Appendix A). This subgroup analysis also showed similar long-term outcomes with 5-year DFSs of 67.6%, 72.9%, and 61.4% in the robotic, laparoscopic, and open groups, respectively, *p* = 0.17 (Appendix A). The 5-year DFS for stage 3 mid and low rectal cancer was 68.1%, 67.7%, and 59.9%, respectively, *p* = 0.49.

## 4. Discussion

There is still a continuous unsolved debate regarding the optimal surgical approach to treat rectal cancer [3,4,5,6]. Therefore, this present study aimed to retrospectively review the short- and long-term outcomes of robotic-assisted approach compared with the laparoscopic-assisted approach and the open approach.

Interestingly, in our study, the robotic group included more patients with distal and clinically locally advanced tumors compared with the other two groups. This was probably the reason for the higher rates of neoadjuvant radiation treatment observed among the robotic group, and also for the higher proportion of abdomino-perineal resections among the robotic group. Nevertheless, those differences did not translate into worse oncological outcomes as could possibly be expected. Similarly, in a study analyzing the short-term outcomes of 2114 consecutive patients in a single center in Korea, the robotic group also had the most distal tumors among the three groups, and the laparoscopic group included less patients with advanced and lower rectal tumors. The authors concluded that laparoscopic-assisted rectal cancer surgery tends to be preferred for upper rectal cancers and less advanced tumors, irrespective of the surgeon’s competence, and that the robotic procedure was probably chosen to overcome surgical complexity in patients with locally advanced and lower rectal cancers [17].

Despite operating on more distal challenging tumors, our robotic group had significantly lower conversion rates to an open procedure compared with the laparoscopic approach. This is in concordance with multiple prior studies that also described significant lower conversion rates among the robotic approach [17,18], while some studies have reported similar conversion rates between the laparoscopic and robotic approach [19,20]. This has clinical significance, as patients converted from the minimally invasive approach to open surgery were found to be at a greater risk for perioperative morbidity and to have worse oncological outcomes [21]. We speculate whether those lower conversion rates among the robotic group are partially related to better patients’ selection, and maybe surgeons using the robotic platform early on in their learning curve have a higher threshold for conversion compared with traditional laparoscopy. In addition, the higher proportion of neoadjuvant radiation among the robotic group, which is known to cause fibrosis and tissue edema that can make surgery technically more difficult [22], has not translated into higher rates of intraoperative complications, nor in intraoperative bleeding, both of which were significantly less prevalent among the robotic approach. Previous meta-analyses and systematic reviews have also observed lower intraoperative blood loss during robotic-assisted rectal cancer surgery compared with laparoscopic and open surgery [20,23]. This could be theoretically explained by the technical advantages of the robotic platform, including three-dimensional high-definition visualization, providing a detailed, stable, and magnified view of the surgical field, instrumental dexterity, and precise and stable dissection, all of which could potentially lead to decreased surgical blood loss, yet we acknowledge the need for cautious interpretation of these results. Still, higher blood loss during surgery has been associated with poor prognosis in colorectal cancer [24]; therefore, these findings suggest that robotic surgery may indirectly improve the prognosis of patients undergoing rectal surgery for cancer.

Consistent with previous studies, we also found a significant shorter hospital stay in the robotic group compared with the other two groups [20,25], and early postoperative complications, specifically ileus and wound infection, occurred more frequently in the open group than in the robotic and laparoscopic groups [17,26,27]. Moreover, major complications (Clavien–Dindo grade > 2) were more prevalent among the open group as well, and this is clinically meaningful as major complications after proctectomy for cancer are associated with earlier disease recurrence, ultimately leading to decreased survival [28]. Our multivariate analysis has found loop ileostomy creation, open surgery, and male gender to be associated with major complications. The latter two factors may be related to limited visibility and a confined narrow workspace, eventually leading to a more technically difficult surgery, possibly dissecting along non-anatomical planes, increasing blood loss, and altogether resulting in an increased postoperative morbidity. In a similar manner, Kim et al. reported that anastomotic complications, including leakage, abscess, fistula, and stricture, were significantly associated with male patients in multivariate analysis (OR, 1.85; 95 % CI, 1.049–3.263; *p* = 0.034), and so performed postoperative ileus (OR, 1.775; 95 % CI, 1.092–2.887; *p* = 0.021) [17]. This strengthens the notion that rectal cancer surgery in male patients is different than in females and could be very challenging and associated with an increased potential for major morbidity. Surgeons and oncologists should consider these issues upon deciding on preoperative treatment, while considering and planning surgery in male patients differently from females. For example, it should be considered to lower the “threshold” for non-operative management in male patients that seem to have a complete clinical response following neoadjuvant therapy.

Our overall postoperative complication rates were relatively high compared with previous studies [23,26]. We believe that this may be related to the definition and classification applied to postoperative morbidity documentation. We followed a very broad interpretation and included in our postoperative complication recordings any deviation from the normal postoperative course (such as electrolyte disturbances).

In line with the previous literature [17,26], the three groups did not differ in the number of retrieved lymph nodes, nor in terms of positive distal/radial margins, both of which serve as benchmarks for surgical oncology quality in rectal cancer surgery. On the other hand, a systematic review by Khajeh et al. found that the robotic approach had significantly higher rates of negative radial margins and a higher number of harvested lymph nodes than the open approach did, and also higher rates of negative radial margins than the laparoscopic approach [23]. The authors stated that this might be related to better visualization and improved access to the pelvis, utilizing the robotic platform. However, the superior histopathological outcomes of the robotic approach have not translated into differences in OS or DFS in this meta-analysis. Our robotic group did show significantly longer distal margins compared with the two other groups, which might also be related to the higher proportions of abdomino-perineal resections performed among this group. Unfortunately, our pathological reports lack reference regarding the completeness of mesorectal excision, which also serves as a surrogate marker for quality rectal cancer surgery.

In this study, the robotic group showed similar long-term oncological outcomes compared with the laparoscopic and open approaches, and there were no significant differences between the groups in overall survival and disease-free survival rates. Those results are similar to previously published studies [23,26,29]. Our multivariate analysis has found shorter tumor distances from the anal verge and involved distal/radial surgical margins to be significantly associated with disease relapse. Similarly, Kim et al. found the local recurrence rates to be closely related to postoperative hemorrhage (OR, 14.02; 95% CI, 2.592–75.84; *p* = 0.002), DRM+ (OR, 13.4; 95% CI, 2.319– 77.439; *p* = 0.004), anastomotic complications (OR, 5.514; 95% CI, 1.416–21.475; *p* = 0.014), and tumor location (OR, 0.364; 95% CI, 0.134–0.989; *p* = 0.047) in multivariate analysis, but their study had a limited follow-up period [17]. The minimal acceptable DRM is controversial, but clearly, a threatened DRM is associated with local recurrence, and approximately three times more local recurrence was reported in patients with DRM <2 cm than in those with DRM >2 cm in one study [30]. This again emphasizes the importance of precise surgical techniques and controlling the distal resection margins in relation to disease-free survival and long-term oncological outcomes. Although we have not found differences in the margin positivity rate, the robotic group in our study presented significantly larger DRMs compared with the other groups, and this can imply that the robotic platform might be better in clear and safe distal resection margins compared with the open and laparoscopic approaches. However, Mirza et al. found that robotic and open TME were associated with higher margin positivity rates (8.2% versus 6.6% versus 1.9%, respectively, *p* = 0.17) compared with laparoscopic TME, and they thought it was related to the higher percentage of low rectal cancers in the robotic and open cohorts [29].

Our study has some limitations. First of all, the non-randomized, retrospective nature of our study is subject to a selection bias due to the surgeon’s preference for each procedure. In addition, this was a single-institution study, which limits the ability to generalize our results. The impact of the learning curve on surgical outcomes, specifically in minimally invasive approaches, such as robotic and laparoscopic, is substantial and should be considered. Our analysis included heterogenous populations of the respective groups regarding tumor location in the rectum, tumor stage, and preoperative radiation therapy, all of which could have an effect on surgical and oncological outcomes. However, this was confronted by subgroup analysis. Despite comprehensive data collection, our report lacks information on the completeness of mesorectal excision, estimated blood loss, and length of surgery, all of which are important markers in comparing the different surgical approaches.

## 5. Conclusions

In summary, our study found no significant differences in pathological outcomes and in long-term oncological outcomes between the robotic, laparoscopic, and open approaches. Additionally, we have demonstrated that minimally invasive approaches may be superior to the open approach in perioperative recovery, and given the lower conversion to open rates of robotic surgery, we believe that the robotic approach should be the preferred approach for rectal cancer, since it allows more patients to benefit from the advantages of the minimally invasive technique.

## Figures and Tables

**Figure 1 cancers-17-00859-f001:**
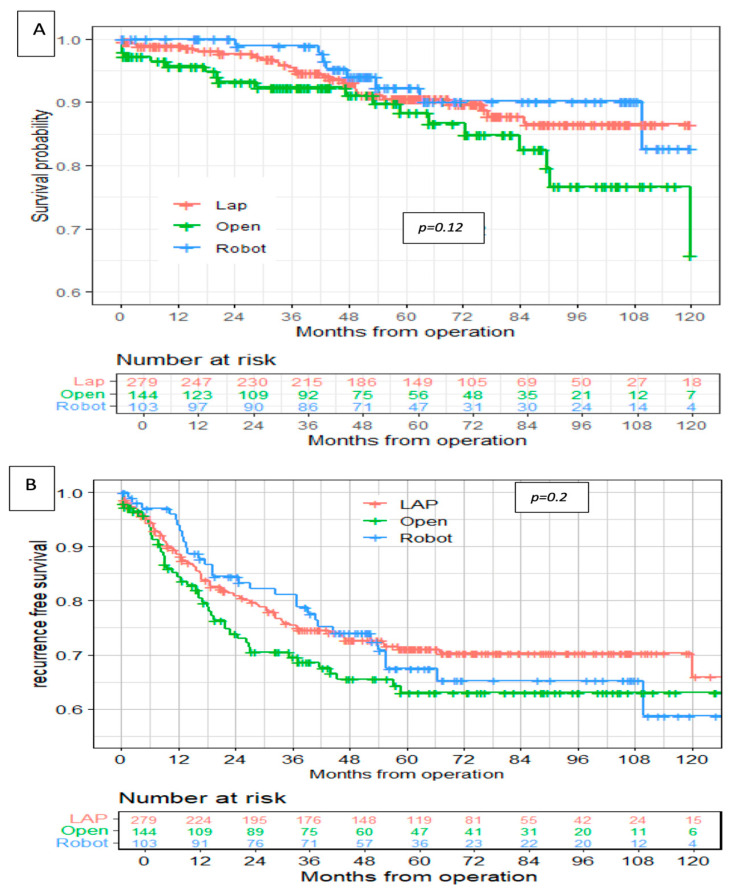
Overall survival (**A**) and recurrence-free survival (**B**).

**Figure 2 cancers-17-00859-f002:**
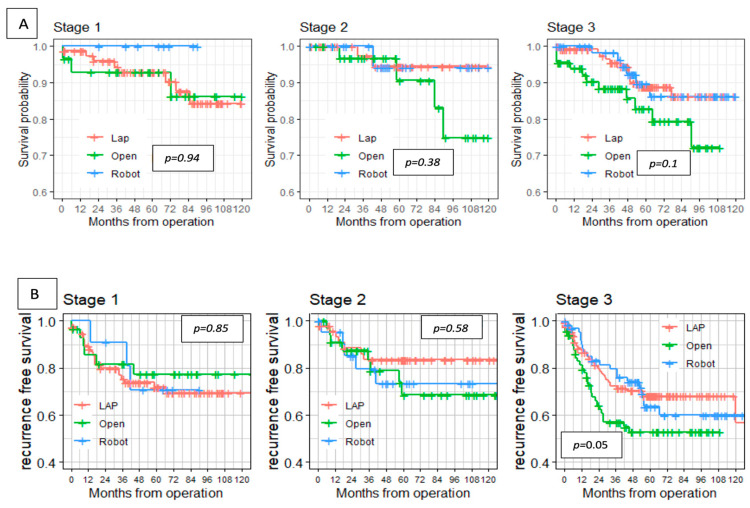
Overall survival (**A**) and recurrence-free survival (**B**) stratified by stage.

**Table 1 cancers-17-00859-t001:** Patients’ characteristics and demographics.

Variable	All Cohort (n = 526)	LAP (n = 279)	Open (n = 144)	Robot (n = 103)	*p*-Value
Age, median (range)	64 (31–89)	64 (31–89)	66 (32–88)	64 (31–83)	0.078
Gender					0.623
Female n (%)	231 (43.9%)	124 (44.4%)	66 (45.8%)	41 (39.8%)	
Male n (%)	295 (56.1%)	155 (55.6%)	78 (54.2%)	62 (60.2%)	
BMI median (range)	26.3 (15.6–45.6)	26.1 (16.8–42.5)	26.9 (15.6–45.6)	25.9 (19.5–41.1)	0.214
Smoking					0.203
No	326 (62%)	179 (64.2%)	90 (62.5%)	57 (55.3%)	
Past smoker	103 (19.6%)	57 (20.4%)	27 (18.8%)	19 (18.4%)	
Current smoker	97 (18.4%)	43 (15.4%)	27 (18.8%)	27 (26.2%)	
Family history of CRC	123 (27.5%)	67 (28.2%)	26 (23%)	30 (31.2%)	0.251
Yes 1st degree	81 (18.1%)	43 (18.1%)	21 (18.6%)	17 (17.7%)	
Yes non-1st degree	42 (9.4%)	24 (10.1%)	5 (4.4%)	13 (13.5%)	
Blood thinners	139 (26.5%)	78 (28%)	40 (27.8%)	21 (20.4)	0.629
Antiaggregating agents	114 (21.7%)	63 (22.6%)	33 (22.9%)	18 (17.5%)	
Anticoagulation	25 (4.8%)	15 (5.4%)	7 (4.9%)	3 (2.9%)	
History of abdominal surgery	212 (40.3)	110 (39.4)	56 (38.9%)	46 (44.7%)	0.282
Prior non-bowel abdominal surgery	154 (29.3%)	83 (29.7%)	35 (24.3%)	36 (35%)	
Prior bowel resection	58 (11%)	27 (9.7%)	21 (14.6%)	10 (9.7%)	
Comorbidity any	415 (78.9%)	220 (78.9%)	119 (82.6%)	76 (73.8%)	0.256
TIA/CVA	18 (3.4%)	6 (2.2%)	10 (6.9%)	2 (1.9%)	**0.024**
Asthma/COPD	48 (9.1%)	28 (10%)	11 (7.6%)	9 (8.7%)	0.712
IHD/CHF	74 (14.1%)	39 (14%)	24 (16.7%)	11 (10.7%)	0.418
Arrhythmia	29 (5.5%)	16 (5.7%)	7 (4.9%)	6 (5.8%)	0.937
DM	117 (22.2%)	56 (20.1%)	37 (25.7%)	24 (23.3%)	0.415
CKD	17 (3.2%)	7 (2.5%)	8 (5.6%)	2 (1.9%)	0.19
HTN	234 (44.5%)	126 (45.2%)	72 (50%)	36 (35%)	0.065
Dyslipidemia	183 (34.8%)	101 (36.2%)	51 (35.4%)	31 (30.1%)	0.535
Hypothyroidism	39 (7.4%)	24 (8.6%)	7 (4.9%)	8 (7.8%)	0.397
ASA score median (range)	3 (1–5)	3 (1–4)	3 (1–5)	3 (1–4)	0.328
ASA score					0.127
1	19 (4.3%)	13 (5.4%)	3 (2.6%)	3 (3.5%)	
2	153 (34.5%)	77 (31.8%)	39 (33.6%)	37 (43%)	
3	257 (57.9%)	148 (61.2%)	67 (57.8%)	42 (48.8%)	
4	14 (3.2%)	4 (1.7%)	6 (5.2%)	4 (4.7%)	

BMI—body mass index, CRC—colorectal cancer, TIA—transient ischemic attack, CVA—cerebrovascular accident, COPD—chronic obstructive pulmonary disease, IHD—ischemic heart disease, CHF—congestive heart failure, DM—diabetes mellitus, CKD—chronic kidney disease, HTN—hypertension, and ASA—American Society of Anesthesiologists, Bold—statistically significant (*p* < 0.05).

**Table 2 cancers-17-00859-t002:** Disease presentation and preoperative work-up.

Variable	All cohort (n = 526)	LAP (n = 279)	Open (n = 144)	Robot (n = 103)	*p*-Value
Pre-diagnosis symptoms	422 (80.2%)	216 (77.4%)	120 (83.3%)	86 (83.5%)	0.236
Abdominal pain	93 (17.7%)	42 (15.1%)	33 (22.9%)	18 (17.5%)	0.137
Anemia	47 (8.9%)	26 (9.3%)	16 (11.1%)	5 (4.9%)	0.231
Weight loss	108 (20.5%)	50 (17.9%)	38 (26.4%)	20 (19.4%)	0.12
Change in bowel movements	222 (42.2%)	119 (42.7%)	64 (44.4%)	39 (37.9%)	0.58
Rectal bleeding	304 (57.8%)	168 (60.2%)	72 (50%)	64 (62.1%)	0.082
Preoperative CT scan	441 (83.8%)	244 (87.5%)	118 (81.9%)	79 (76.7%)	0.031
Preoperative PET scan	203 (38.6%)	95 (34.1%)	57 (39.6%)	51 (49.5%)	0.022
Preoperative TRUS	362 (68.8%)	215 (77.1%)	90 (62.5%)	57 (55.3%)	<0.001
Preoperative pelvic MRI	260 (49.4%)	153 (54.8%)	69 (47.9%)	38 (36.9%)	0.037
Colonoscopy	522 (99.2%)	278 (99.6%)	142 (98.6%)	102 (99%)	0.694
Tumor location					**<0.001**
Lower rectum (<5 cm)	60 (11.4%)	17 (6.1%)	18 (12.5%)	25 (24.3%)	
Mid rectum (5 ≤ X ≤ 10 cm)	169 (32.1%)	76 (27.2%)	55 (38.2%)	38 (36.9%)	
Upper rectum (>10 cm)	297 (56.5%)	186 (66.7%)	71 (49.3%)	40 (38.8%)	
Distance from anal verge, median (range)	10 (0–15)	12 (0–15)	9 (1–15)	8 (0–15)	**<0.001**
Clinical stage					**0.003**
Stage 1	119 (24.6%)	79 (30.7%)	29 (22.1%)	11 (11.5%)	
Stage 2	106 (21.9%)	49 (19.1%)	35 (26.7%)	22 (22.9%)	
Stage 3	259 (53.5%)	129 (50.2%)	67 (51.1%)	63 (65.6%)	
Clinical T stage					**<0.001**
T1	47 (9.7%)	32 (12.5%)	10 (7.6%)	5 (5.2%)	
T2	101 (20.9%)	65 (25.3%)	25 (19.1%)	11 (11.5%)	
T3	300 (62%)	155 (60.3%)	74 (56.5%)	71 (74%)	
T4	36 (7.4%)	5 (1.9%)	22 (16.8%)	9 (9.4%)	
Clinical N stage					0.108
N0	235 (47.6%)	134 (51%)	65 (49.2%)	36 (36.4%)	
N1	187 (37.9%)	89 (33.8%)	51 (38.6%)	47 (47.5%)	
N2	72 (14.6%)	40 (15.2%)	16 (12.1%)	16 (16.2%)	
Clinical EMVI	9 (4%)	4 (3.2%)	1 (1.9%)	4 (8.9%)	0.151
Preoperative Albumin, median g/dL (range)	4.1 (2.4–5.1)	4.2 (2.5–5.1)	4 (2.4–4.9)	4.1 (2.5–4.8)	0.06
Preoperative Hgb, median g/dL (range)	12.7 (7.8–17.2)	12.8 (8.7–17.2)	12.5 (7.8–16)	12.8 (9–16)	0.272
Preoperative Creatinine, median (range)	0.8 (0.2–2.5)	0.8 (0.4–2.5)	0.8 (0.2–1.9)	0.8 (0.4–1.4)	0.986
Preop CEA, median (range)	2 (0–175)	1.8 (0–175)	2.2 (0–140.9)	2 (0–78.3)	0.057
Preop CA 19-9, median (range)	9.3 (0–248)	8.9 (0–248)	10.8 (0–188)	9.2 (0–92.1)	0.077
Neoadjuvant radiation	261 (49.7%)	109 (39.1%)	79 (54.8%)	73 (70.9%)	**<0.001**
Short-course radiotherapy	34 (6.5%)	10 (3.6%)	11 (7.6%)	13 (12.6%)	
Long-course chemo-radiation	227 (43.2%)	99 (35.5%)	68 (47.2%)	60 (58.3%)	

TRUS—transrectal ultrasound, EMVI—extramural vascular invasion, Hgb—hemoglobin, and CEA—carcinoembryonic antigen, Bold—statistically significant (*p* < 0.05).

**Table 3 cancers-17-00859-t003:** Operative details.

Variable	All Cohort (n = 526)	LAP (n = 279)	Open (n = 144)	Robot (n = 103)	*p*-Value
Procedure					**<0.001**
Abdomino-perineal resection	33 (6.3%)	12 (4.3%)	3 (2.1%)	18 (17.6%)	
Anterior resection	175 (33.4%)	113 (40.6%)	43 (29.9%)	19 (18.6%)	
Low anterior resection (TME)	316 (60.3%)	153 (55%)	98 (68.1%)	65 (63.7%)	
Stoma created	321 (61%)	146 (52.3%)	98 (68%)	77 (74.7)	**<0.001**
Diverting loop ileostomy	269 (51.1%)	124 (44.4%)	87 (60.4%)	58 (56.3%)	
End colostomy	52 (9.9%)	22 (7.9%)	11 (7.6%)	19 (18.4%)	
Stoma reversed	222 (69.2%)	104 (71.2%)	68 (69.4%)	50 (64.9%)	0.64
Additional procedure	81 (15.4%)	27 (9.7%)	46 (31.9%)	8 (7.8%)	**<0.001**
Additional anastomosis	17 (3.2%)	5 (1.8%)	12 (8.3%)	0 (0%)	0.001
BSO	40 (7.6%)	10 (3.6%)	28 (19.4%)	2 (1.9%)	**<0.001**
Hysterectomy	10 (1.9%)	3 (1.1%)	6 (4.2%)	1 (1%)	0.077
Hernia repair	6 (1.1%)	2 (0.7%)	2 (1.4%)	2 (1.9%)	0.646
Cholecystectomy	3 (0.6%)	2 (0.7%)	1 (0.7%)	0 (0%)	0.829
Type of anastomosis					<0.001
Colonic pouch	13 (2.7%)	4 (1.6%)	5 (3.7%)	4 (4.8%)	
End-to-end	362 (76.2%)	215 (83.7%)	72 (53.7%)	75 (89.3%)	
Hand-sewn coloanal	31 (6.5%)	11 (4.3%)	17 (12.7%)	3 (3.6%)	
Side-to-end	69 (14.5%)	27 (10.5%)	40 (29.9%)	2 (2.4%)	
Skin incision					0.001
Left lower quadrant	7 (1.8%)	7 (2.5%)	-	0 (0%)	
Midline laparotomy	66 (17.3%)	60 (21.5%)	-	6 (5.8%)	
Natural orifice	50 (13.1%)	30 (10.8%)	-	20 (19.4%)	
Periumbilical	20 (5.2%)	16 (5.7%)	-	4 (3.9%)	
Pfanensteil	239 (62.6%)	166 (59.5%)	-	73 (70.9%)	
Conversion	70 (18.3%)	63 (22.6%)	-	7 (6.8%)	**0.001**
Intraoperative complication	82 (15.6%)	38 (13.6%)	33 (22.9%)	11 (10.7%)	**0.015**
Bleeding	27 (5.1%)	9 (3.2%)	15 (10.4%)	3 (2.9%)	**0.004**
Ureter/Urethra injury	4 (0.8%)	2 (0.7%)	2 (1.4%)	0 (0%)	0.443
Enterotomy	23 (4.4%)	10 (3.6%)	11 (7.6%)	2 (1.9%)	0.065
Splenic injury	7 (1.3%)	2 (0.7%)	5 (3.5%)	0 (0%)	0.018
Vaginal injury	8 (1.5%)	3 (1.1%)	1 (0.7%)	4 (3.9%)	0.105
Anastomosis disruption	8 (1.5%)	6 (2.2%)	1 (0.7%)	1 (1%)	0.566

TME—total mesorectal excision; BSO—bilateral salpingo-ophorectomy, Bold—statistically significant (*p* < 0.05).

**Table 4 cancers-17-00859-t004:** Postoperative immediate surgical outcomes.

Variable	All Cohort (n = 526)	LAP (n = 279)	Open (n = 144)	Robot (n = 103)	*p*-Value
LOS, days, median (range)	8 (3–98)	8 (3–86)	10 (4–98)	7 (4–53)	<0.001
Postoperative complications, overall	345 (65.6%)	164 (58.8%)	110 (76.4%)	71 (68.9%)	0.001
SSI	78 (14.8%)	29 (10.4%)	41 (28.5%)	8 (7.8%)	<0.001
Intra-abdominal abscess	37 (7%)	20 (7.2%)	11 (7.6%)	6 (5.8%)	0.854
Ileus/SBO	113 (21.5%)	53 (19%)	44 (30.6%)	16 (15.5%)	0.007
Anastomotic leak	44 (8.4%)	20 (7.2%)	18 (12.5%)	6 (5.8%)	0.106
Bleeding	74 (14.1%)	38 (13.6%)	26 (18.1%)	10 (9.7%)	0.173
Pneumonia	12 (2.3%)	6 (2.2%)	3 (2.1%)	3 (2.9%)	0.927
UTI	33 (6.3%)	12 (4.3%)	15 (10.4%)	6 (5.8%)	0.053
DVT	3 (0.6%)	1 (0.4%)	2 (1.4%)	0 (0%)	0.43
MI/arrhythmia	19 (3.6%)	6 (2.2%)	9 (6.2%)	4 (3.9%)	0.105
Wound dehiscence	10 (1.9%)	3 (1.1%)	5 (3.5%)	2 (1.9%)	0.251
Electrolyte disturbances/ARF	182 (34.6%)	79 (28.3%)	58 (40.3%)	45 (43.7%)	0.006
Urinary retention	38 (7.2%)	17 (6.1%)	8 (5.6%)	13 (12.6%)	0.064
Clavien–Dindo score, median (range)	1 (0–5)	1 (0–5)	1 (0–5)	1 (0–4)	<0.001
Major complications (Clavien–Dindo > 2)	84 (16%)	35 (12.5%)	35 (24.3%)	14 (13.6%)	0.007
Surgery-related 30d readmission	92 (17.5%)	43 (15.4%)	32 (22.2%)	17 (16.5%)	0.215
30d mortality	6 (1.1%)	2 (0.7%)	4 (2.8%)	0 (0%)	0.074

LOS—length of stay, SSI—surgical site infection, SBO—small bowel obstruction, UTI—urinary tract infection, DVT—deep vein thrombosis, and ARF—acute renal failure.

**Table 5 cancers-17-00859-t005:** Univariate (A) and multivariate (B) logistic regression model for variables associated with major postoperative complications.

Variable	OR	Lower CI	Upper CI	Pr(>|z|)	*p*-Value
A. Univariate analysis					
Age	1.0114	0.9913	1.0318	0.2694	0.2658
Gender male	1.6987	1.0400	2.7749	0.0343	**0.0312**
BMI <= 20	1				
BMI 21–25	1.3500	0.3761	4.8456	0.6453	0.9628
BMI 26–30	1.3462	0.3808	4.7584	0.6445	
BMI >= 31	1.4177	0.3773	5.3268	0.6053	
No smoking	1				
Past smoker	0.8586	0.4616	1.5969	0.6301	0.8784
Active smoker	0.9214	0.4941	1.7181	0.7968	
Tumor distance from AV	0.9153	0.8652	0.9683	0.0021	**0.0020**
Surgical approach: LAP	1				
Open approach	2.2385	1.3307	3.7657	0.0024	**0.0079**
Robotic approach	1.0966	0.5636	2.1337	0.7859	
No preoperative radiation	1				
Preop RAD short course	1.9573	0.8226	4.6572	0.1289	0.2322
Preop RAD long course	1.3607	0.8336	2.2210	0.2180	
No stoma	1				
Stoma DLI	3.1056	1.7231	5.5976	0.0002	0.0001
Stoma end colostomy	3.5437	1.5568	8.0667	0.0026	
B. Multivariate analysis					
Age	1.006	0.984	1.029	0.607	0.606
Gender male	1.676	1.004	2.798	0.048	**0.048**
Tumor distance from AV	0.957	0.880	1.042	0.310	0.309
Open approach	1.959	1.132	3.390	0.016	**0.026**
Robotic approach	0.897	0.441	1.824	0.764	
Preop RAD short course	0.697	0.256	1.899	0.480	0.155
Preop RAD long course	0.515	0.262	1.011	0.054	
Stoma DLI	3.416	1.570	7.431	0.002	**0.007**
Stoma end colostomy	3.911	1.295	11.815	0.016	

OR—odds ratio, CI—confidence interval, BMI—body mass index, AV—anal verge, LAP—laparoscopy, RAD—radiation, and DLI—diverting loop ileostomy. Pr(>|z|)—significance (relative to reference category), Bold—statistically significant (*p* < 0.05).

**Table 6 cancers-17-00859-t006:** Pathological results.

Variable	All Cohort (n = 526)	LAP (n = 279)	Open (n = 144)	Robot (n = 103)	*p*-Value
Complete pathological response	54 (10.4%)	32 (11.6%)	11 (7.9%)	11 (10.7%)	0.512
Tumor size, cm, median (range)	2.8 (0–11)	2.8 (0–11)	3 (0.1–8)	2.2 (0.2–10.2)	0.054
Tumor differentiation					0.993
Well differentiated	135 (36%)	72 (35.5%)	38 (35.8%)	25 (37.9%)	
Moderately differentiated	219 (58.4%)	119 (58.6%)	62 (58.5%)	38 (57.6%)	
Poorly differentiated	21 (5.6%)	12 (5.9%)	6 (5.7%)	3 (4.5%)	
Pathological stage					0.57
0	54 (10.4%)	32 (11.5%)	11 (7.9%)	11 (10.8%)	
I	173 (33.3%)	99 (35.6%)	42 (30%)	32 (31.4%)	
II	126 (24.2%)	58 (20.9%)	39 (27.9%)	29 (28.4%)	
III	161 (31%)	85 (30.6%)	46 (32.9%)	30 (29.4%)	
IV	6 (1.2%)	4 (1.4%)	2 (1.4%)	0 (0%)	
Pathological T stage					0.085
T0	55 (10.6%)	33 (11.9%)	11 (7.9%)	11 (10.8%)	
T1	62 (12%)	42 (15.2%)	10 (7.2%)	10 (9.8%)	
T2	139 (26.8%)	73 (26.4%)	40 (28.8%)	26 (25.5%)	
T3	252 (48.6%)	126 (45.5%)	72 (51.8%)	54 (52.9%)	
T4	10 (1.9%)	3 (1.1%)	6 (4.3%)	1 (1%)	
Lymph nodes harvested, n (range)	14 (0–113)	14 (0–113)	16 (2–65)	14 (0–54)	0.558
Pathological N stage					0.755
N0	353 (67.9%)	190 (68.3%)	92 (65.7%)	71 (69.6%)	
N1a (<2 LNs)	45 (8.7%)	25 (9%)	15 (10.7%)	5 (4.9%)	
N1b (2–3 LNs)	49 (9.4%)	25 (9%)	14 (10%)	10 (9.8%)	
N1c	39 (7.5%)	18 (6.5%)	11 (7.9%)	10 (9.8%)	
N2a (4–6 LNs)	18 (3.5%)	13 (4.7%)	3 (2.1%)	2 (2%)	
N2b (>7 LNs)	14 (2.7%)	6 (2.2%)	5 (3.6%)	3 (2.9%)	
Lympho-vascular invasion	39 (7.4%)	18 (6.5%)	15 (10.4%)	6 (5.8%)	0.271
Perineural invasion	40 (7.6%)	22 (7.9%)	11 (7.6%)	7 (6.8%)	0.954
Distal margin, cm, median (range)	2.5 (0.1–9)	2.5 (0.1–9)	1.9 (0.2–5.5)	3.5 (0.3–7)	**0.008**
Involved radial/distal margins	11 (2.1%)	5 (1.8%)	5 (3.5%)	1 (1%)	0.373
Signet ring	11 (2.1%)	3 (1.1%)	4 (2.8%)	4 (3.9%)	0.193
Mucinous tumors	88 (16.7%)	42 (15.1%)	26 (18.1%)	20 (19.4%)	0.533

Bold—statistically significant (*p* < 0.05).

**Table 7 cancers-17-00859-t007:** Long-term oncological outcomes.

Variable	All Cohort (n = 526)	LAP (n = 279)	Open (n = 144)	Robot (n = 103)	*p*-Value
Median follow-up time, months (range)	59 (1–171)	61.6 (1–171.4)	50 (1–153)	56 (1–143)	0.05
Adjuvant chemotherapy	252 (47.9%)	130 (46.6%)	60 (41.7%)	62 (60.2%)	**0.014**
Disease recurrence	132 (25.1%)	64 (22.9%)	39 (27.1%)	29 (28.2%)	0.279
Local recurrence	43 (8.2%)	19 (6.8%)	17 (11.8%)	7 (6.8%)	
Distant recurrence	89 (16.9%)	45 (16.1%)	22 (15.3%)	22 (21.4%)	
Overall mortality during follow-up time	52 (9.9%)	25 (9%)	19 (13.2%)	8 (7.8%)	0.288

Bold—statistically significant (*p* < 0.05).

**Table 8 cancers-17-00859-t008:** Cox regression model for risk factors associated with disease recurrence.

Variable	HR	Lower CI	Upper CI	Pr(>|z|)	*p*-Value
A. Univariate analysis					
Age	1.005	0.990	1.020	0.499	0.497
Gender male	1.046	0.740	1.477	0.800	0.800
BMI 21–25	1.005	0.431	2.343	0.990	0.171
BMI 26–30	0.646	0.276	1.515	0.315	
BMI >= 31	0.855	0.351	2.083	0.730	
Past smoker	0.925	0.595	1.436	0.727	0.596
Active smoker	0.789	0.493	1.262	0.323	
Tumor distance from AV	0.951	0.912	0.991	0.017	**0.017**
Surgical approach: LAP	1				
Open approach	1.305	0.876	1.944	0.190	0.411
Robotic approach	1.174	0.757	1.822	0.473	
Major complications	1.212	0.767	1.915	0.411	0.421
Positive pathological margins	3.675	1.357	9.952	0.010	**0.034**
Positive lymph nodes	1.000	0.985	1.015	0.971	0.971
B. Multivariate analysis					
Age	1.006	0.991	1.021	0.460	0.458
Tumor distance from AV	0.954	0.914	0.996	0.034	**0.034**
Open approach	1.199	0.800	1.797	0.379	0.682
Robotic approach	1.062	0.669	1.684	0.800	
Positive pathological margins	3.599	1.320	9.812	0.012	**0.037**

Bold—statistically significant (*p* < 0.05).

## Data Availability

The data presented in this study are available upon request from the corresponding author due to privacy and ethical reasons.

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
