# Peer review of "Robotic Rectal Cancer Surgery: Perioperative and Long-Term Oncological Outcomes of a Single-Center Analysis Compared with Laparoscopic and Open Approach"

_cancers, 2025, doi:10.3390/cancers17050859_

Round 1

Reviewer 1 Report

Comments and Suggestions for Authors

Laks et al. have compared the short and long-term outcomes of robotic assisted approach with laparoscopic assisted and open approaches in patients undergone elective rectal resection for primary non-metastatic rectal cancer. The cohort included 526 patients (103 patients were in the robotic group. In this study,  robotic, laparoscopic and open approaches had similar histopathological outcomes and long-term oncological outcomes.

The study is interesting and clinically important, especially cause open surgery is still a frequent method. I have following comments:

1)     Please ad the country of investigation to methods of the abstract.

2)     Please describe in method text details of the logistic regression. Multivariable logistic regression was adjusted for different covariables. Authors should explain which variables were included in the model to adjust for.

3)     Authors should state in methods, which p value was considered statistically significant (<0.05?)

4)     Table 5- what does “na” in the p-valu column mean?

5)     What is “Pr(>|z|)”, is it needed in the table ?

6)     Table 6 contains open and robotic but not laparoscopic approach. Has laparoscopic been used a a reference group? In this case, that should be mentioned/shown.

7)     There is another reference what may be important but is missing in your work: doi: 10.1007/s00464-024-11210-1

Reviewer 2 Report

Comments and Suggestions for Authors

·         The authors conclude that the robotic approach is safe and with similar results compared to lap and open. In my opinion having similar preoperative and oncological long-term outcomes between a group of robotic cases including more locally advanced, low rectal tumours and a group of lap cases including more early upper rectal cancers emphasizes, unproven yet, the superiority of the robotic approach. For early, upper rectal cancers, any approach will provide good quality specimen, good margins and excellent survival rates. Where the robotic platform brings an undeniable advantage is within the deep narrow pelvis, for a low rectal tumour, where one aims to perform an anterior resection rather than APR (considering that for a very low rectal cancer, the margins are dealt with largely through the open perineal approach regardless of the type of abdominal approach). And this brings me to the next comment.

·         All results are faulted and biased due to important confounders in the three groups. There are more locally advanced, low rectal cancers with neoadjuvant therapy in robotic group. This confounding bias needs to be dealt with by the authors either by propensity score matching or by simply performing a similar subgroup analysis including only low rectal cancers, that underwent neoadjuvant therapy. Another subgroup analysis could be made on mid rectal cancers, where a low anterior resection would be performed. I predict the robotic approach will be superior to lap, but similar to open.  Whereas, for upper rectal cancer, I believe all three are similar.

·         I would also like to see the DFS rates only in mid-low stage III rectal cancers between the three approaches.

Round 2

Reviewer 1 Report

Comments and Suggestions for Authors

-